

# The effects of exercise on oxidative stress MDA and SOD in patients with type 2 diabetes: a systematic review and meta-analysis

Chen Qiu[1], Shufan Li[1], Shuqi Jia[1], Fen Yu[1], Chen Wei[2], Xing Wang[1] and Bo Guo[3]

[1] Shanghai University of Sport, Shanghai, China
[2] Shenyang Normal University, Shenyang, China
[3] Binzhou Medical University, Binzhou, China

## ABSTRACT

**Objective:** Systematic review of the effect of exercise intervention on oxidative stress in patients with type 2 diabetes mellitus (T2DM).

**Methods:** Databases such as Embase, Web of Science, PubMed, The Cochrane Library, Wanfang, VIP, and China National Knowledge Infrastructure were searched from their inception to June 2024. PEDro was used to assess the quality of the literature, RevMan 5.4.1 and Stata 17.0 were used to perform meta-analysis and publication bias tests, respectively, and GRADEPro was used to evaluate the quality of the evidence for outcome indicators. Standardized mean difference (SMD) and 95% confidence interval (CI) were used as effect measures.

**Results:** This study included 11 articles (1,111 patients). The results of the meta-analysis showed that exercise can improve malondialdehyde (MDA), standardized mean difference (SMD) = −1.29, 95% CI [−1.87 to −0.71], $P < 0.0001$ in patients with T2DM; and improve superoxide dismutase (SOD), SMD = 0.59, 95% CI [0.17–1.01], $P = 0.006$ in patients with T2DM. Subgroup analysis showed that exercise can improve MDA and SOD in patients aged >60 years. The effect is significant when the intervention method is aerobic exercise and combined exercise. The intervention period should be >12 weeks and intervention frequency <3 days/week. Exercise is more effective in improving MDA, an indicator of oxidative stress, when the intervention method is aerobic exercise, the period is >12 weeks and the frequency is <5 days/week, which improves oxidative stress indicators SOD.

**Discussion:** This study included 11 articles with an average PEDro score of 6.5, indicating good quality. Subgroup and sensitivity analyses of the oxidative stress indicators MDA and SOD failed to identify sources of heterogeneity, which is a limitation. Publication bias tests for the oxidative stress indicators MDA and SOD suggest that there is no significant publication bias. Therefore, moderate-quality evidence is given to the oxidative stress indicator MDA, and high-quality evidence is given to the oxidative stress indicator SOD. Exercise has a significant effect on improving oxidative stress indicators MDA and SOD in patients with type 2 diabetes. It is affected by the patient's age, the method, duration, and frequency of the intervention. It can provide evidence-based medical evidence for the clinical rehabilitation of patients with type 2 diabetes.

Corresponding authors
Xing Wang, 18930132117@163.com
Bo Guo, guoboren2002@163.com

# INTRODUCTION

Type 2 diabetes mellitus (T2DM) is a chronic metabolic disease characterized primarily by insulin resistance and progressive decline in pancreatic β-cell function. It is the most common form of diabetes (*Wen et al., 2026*). It is characterized by insulin resistance, lipid accumulation, impaired glycogen synthesis, and mitochondrial function, which prevents glucose from being transported to the body's cells for storage in response to insulin stimulation (*Hong et al., 2024*). The pathogenesis of type 2 diabetes is caused by the interaction of insulin dysfunction and insulin resistance (*Wang, 2019*). Currently, approximately 589 million people worldwide are living with diabetes, with China leading globally with 148 million cases (*Liu, 2025*). The incidence of diabetes is rising worldwide, posing a huge risk to the health and lives of patients and presenting a heavy economic burden (*Ogurtsova et al., 2017*). Therefore, preventing and alleviating the development of type 2 diabetes is an urgent issue that needs to be addressed.

Oxidative stress refers to an imbalance between the production of reactive oxygen species (ROS) and reactive nitrogen species (RNS) and the removal of these substances by the body's antioxidant defense system. This produces excessive ROS and RNS production, which damage body tissues, cells, proteins, and other biological macromolecules such as nucleic acids (*Reddy et al., 2009*). The most representative of these is the increase in the concentration of the oxidative stress marker malondialdehyde (MDA) and the decrease in the concentration of superoxide dismutase (SOD)(*Ogurtsova et al., 2017*; *Yan et al., 2020*), which leads to damage to the function of pancreatic beta cells and peripheral insulin resistance, which induces the formation of diabetes. In severe cases, it can lead to the formation of various complications, such as diabetic neuropathy (*Hernández-Beltrán, Moreno & Gutiérrez-Álvarez, 2013*; *Souza et al., 2011*), diabetic retinopathy (*Selvaraju et al., 2012*), and diabetic cardiovascular disease (*Ren et al., 2013*; *Zhou, 2020*; *Wang et al., 2020*). It is an important factor in developing type 2 diabetes (*Zhao, Lv & Cheng, 2013*). Studying the role of oxidative stress in the pathogenesis of type 2 diabetes not only provides a more comprehensive understanding of the pathogenesis of type 2 diabetes but also provides a theoretical basis for the prevention and treatment of type 2 diabetes (*Ruan et al., 2023*).

Many pharmacological treatments for type 2 diabetes have been reported (*Chaudhury et al., 2017*), but the cost of medication is accompanied by numerous side effects, related complications (*Stumvoll & Häring, 2002*), and healthcare expenses, which can become a burden for T2DM. Moreover, relevant studies have found that functional foods (such as pomegranate) have certain beneficial effects on oxidative stress associated with type 2 diabetes. However, the evidence supporting the role of pomegranate in alleviating oxidative stress primarily comes from preclinical studies, which are often difficult to translate into clinical settings (*Mokgaloboni et al., 2023*). Exercise as an effective treatment has the advantages of being low-cost, easy to operate and organize. In recent years, it has been widely used in the rehabilitation of type 2 diabetes and is one of the main forms of

lifestyle intervention for T2DM patients (*Wang, 2019*). Reviewing previous research, *Venugopal et al. (2022)* believe that yoga exercise can improve oxidative stress markers MDA and blood sugar in type 2 diabetes patients but does not affect the oxidative stress marker SOD. *Guo, Bo & Peng (2014)* believe that exercise has a beneficial effect on the oxidative stress marker SOD in patients with type 2 diabetes (*Hegde et al., 2020*; *Guo, Bo & Peng, 2014*; *Zhang et al., 2012*; *Promsrisuk et al., 2023*; *Hegde et al., 2011*; *Li, 2013*; *Wang et al., 2015*; *Wei et al., 2023*; *Gordon et al., 2008*; *Vasconcelos Gouveia et al., 2021*; *Xu, Jing & Zhao, 2019*). Related articles have shown that aerobic exercise such as walking, running, and cycling can significantly increase the levels of antioxidant enzymes, especially the activity of SOD and glutathione peroxidase (GPx), and reduce the concentration of lipid peroxidation products MDA, as well as reducing oxidative damage (*Stambler et al., 2005*; *Chen & Sun, 2023*). In addition, resistance training can effectively reduce oxidative stress levels by increasing muscle mass and enhancing metabolic function (*Wang, Chang & Wang, 2022*). The combined aerobic and resistance training intervention has a better effect, improving the state of oxidative stress and enhancing insulin sensitivity (*Li et al., 2024*) and blood glucose control (*Kanaley et al., 2022*), and has become an important part of T2DM management. The efficacy of exercise interventions is gradually being recognized compared to conventional rehabilitation treatments (*Joslin, 1916*).

Through a review of the literature, we found that previous articles have been controversial about whether exercise can improve MDA and SOD, markers of oxidative stress, in patients with type 2 diabetes. Exercise regimens were not precise enough, and it was not clear whether the changes in MDA and SOD, markers of oxidative stress, in patients with type 2 diabetes were affected by exercise elements such as exercise intervention methods, intervention cycles, and intervention frequencies. This study aimed to use a systematic review to explore the intervention effect of different exercise modes on oxidative stress in patients with type 2 diabetes. On this basis, it further clarifies whether the intervention effect is affected by the patient's age and various exercise elements to formulate a precise exercise plan for the exercise intervention of patients with type 2 diabetes, provide a scientific basis for preventing and improving oxidative stress damage in patients with type 2 diabetes, and providing a reference for clinical practice.

## SURVEY METHODOLOGY

This study followed the requirements of the PRISMA statement for reporting on meta-analyses (*Page et al., 2021*) for the selection and use of research methods and has been registered in the International Prospective Register of Systematic Reviews (PROSPERO) (CRD42024593503).

### Research framework

See Table S1.

### Search strategy

Computer searches were performed on CNKI, WanFang, VIP, PubMed, Embase, Cochrane Library, and Web of Science, and all searches were performed from the time of
construction to 30, June, 2024. The search strategy was based on the combination of subject terms with free words, with 'Oxidative Stress/Oxidative Injury/Antioxidative', 'Sport/exercise/walking/Tai Chi/yoga/swimming/Anaerobic exercise/Strength Training/ Resistance Training/physical activity/Pilates/I Ching/dance', "type 2 diabetes Mellitus", and 'type 2 diabetes Mellitus/Noninsulin-Dependent Diabetes Mellitus/diabetes type II/ T2DM', "Randomized controlled trial/Randomized" for 'Randomised controlled trial/ Randomised' were the search terms.

After retrieving the relevant literature from each database, the literature was imported into the Endnote X9 software for weight removal. The two authors, Fen Yu and Shuqi Jia, screened the literature according to the inclusion and exclusion criteria in an independent double-blind manner, reading the titles and abstracts for preliminary elimination of the articles, and then downloading the full text of the articles after obtaining the qualified literature and performing full-text screening. At the end of screening, the two authors compared the extracted literature, and in case of disagreement, they consulted with the 3rd author, Shufen Li, until a consensus was reached. The literature search strategy is shown in Table S2.

## Literature inclusion and exclusion criteria
### Inclusion criteria
(1) The research subjects were patients with type 2 diabetes (T2MDs' age ≥ 45 years old), and the diagnostic criteria for T2MD were as follows: in line with the 1999 WHO criteria of fasting plasma glucose (FPG) >7.0 mmol/L, 2 h post-load plasma glucose (2 h PG) ≥11.1 mmol/LJ, and no organic lesions (*Society, 2018*). Type 1 diabetes patients, malnutrition, severe type 2 diabetes complications, and severe cardiovascular disease patients were excluded. (2) The intervention was exercise or exercise based on the control group, and the intervention period was ≥4 weeks. (3) The control group received routine treatment (including drug therapy). (4) The main outcome indicators were oxidative stress indicators MDA and SOD. (5) The study type was randomized controlled trial (RCT). (6) No language restrictions were applied.

### Exclusion criteria
(1) Duplicate publications. (2) Experimental data that could not be calculated or extracted. (3) The control group had exercise intervention. (4) The control group was a healthy population. (5) The intervention method did not match, and the outcome indicators did not match. (6) The full text could not be obtained .rs MDA and SOD. (5) The study type was RCT.

## Literature screening, data extraction, and quality evaluation
### Literature screening and data extraction
After retrieving the relevant literature, the literature was imported into Endnote X9 (*Yue, Zhong & Wang, 2014*) for deduplication. Two researchers independently screened the literature and extracted the data. The data extracted from the eligible articles were entered into RevMan 5.4.1 and double-checked for accuracy. If there were a disagreement,

a third researcher would join the discussion to decide whether to include it. The author's name, year of publication, country of origin, baseline characteristics of the study population (age), intervention strategies, and outcome measures, among other relevant information.

### Quality evaluation

The methodological quality of the included literature was evaluated using the PEDro scale (*Ludyga et al., 2020*). The scale includes ten items such as 'eligibility criteria,' 'random allocation,' 'allocation concealment,' 'baseline similarity,' 'blinding of participants', 'blinding of therapists', 'blinding of outcome assessment,' 'participation rate > 85%', 'intention-to-treat analysis,' 'analysis of statistical results between groups', and 'point measurement difference value.' One point is awarded for meeting a certain criterion, and zero points are awarded for not meeting it. The total score on the scale is 10 points. A score of <4 is considered low quality, 4–5 is considered moderate quality, 6–8 is considered relatively good quality, and 9–10 is considered high quality. Only literature with moderate quality or above is included in this article. The quality of the included studies was assessed independently by two reviewers. In case of any discrepancies, a third reviewer was consulted to resolve disagreements through discussion until consensus was reached.

The GRADEpro evidence grading system (*Yao et al., 2014*) was also used to evaluate the quality of the evidence for outcome indicators, which were divided into four categories from high to low: high, moderate, low, and very low. Two researchers independently assessed the quality scores. If there was a difference, a third researcher was involved in the discussion until a consensus was reached.

## Data processing

RevMan 5.4.1 software was used to combine effect sizes, test for heterogeneity, and perform subgroup analyses for the outcome indicators of the included literature. Stata 17.0 was used to perform sensitivity analyses and publication bias tests. The outcome indicators included were all continuous variables. The mean difference (MD) was used for outcome indicators with the same measurement method and unit, and the standardized mean difference (SMD) was used for outcome indicators with different measurement methods or units. The units of measurement for oxidative stress were not completely the same in each study, so SMD was used. The $P$ value and $I^2$ were used to test for heterogeneity. $P < 0.05$ and $I^2 > 50\%$ indicated heterogeneity among articles, and the random effects model was used; otherwise, there was no significant heterogeneity among articles, and the fixed effects model was used. In this study, both outcome indicators, MDA and SOD, were analyzed using a random-effects model.

## RESULTS

### Results of the literature search

A preliminary search yielded 2,727 documents. After removing duplicates, 463 documents were screened. After reading the titles and abstracts, 2,264 documents were eliminated as
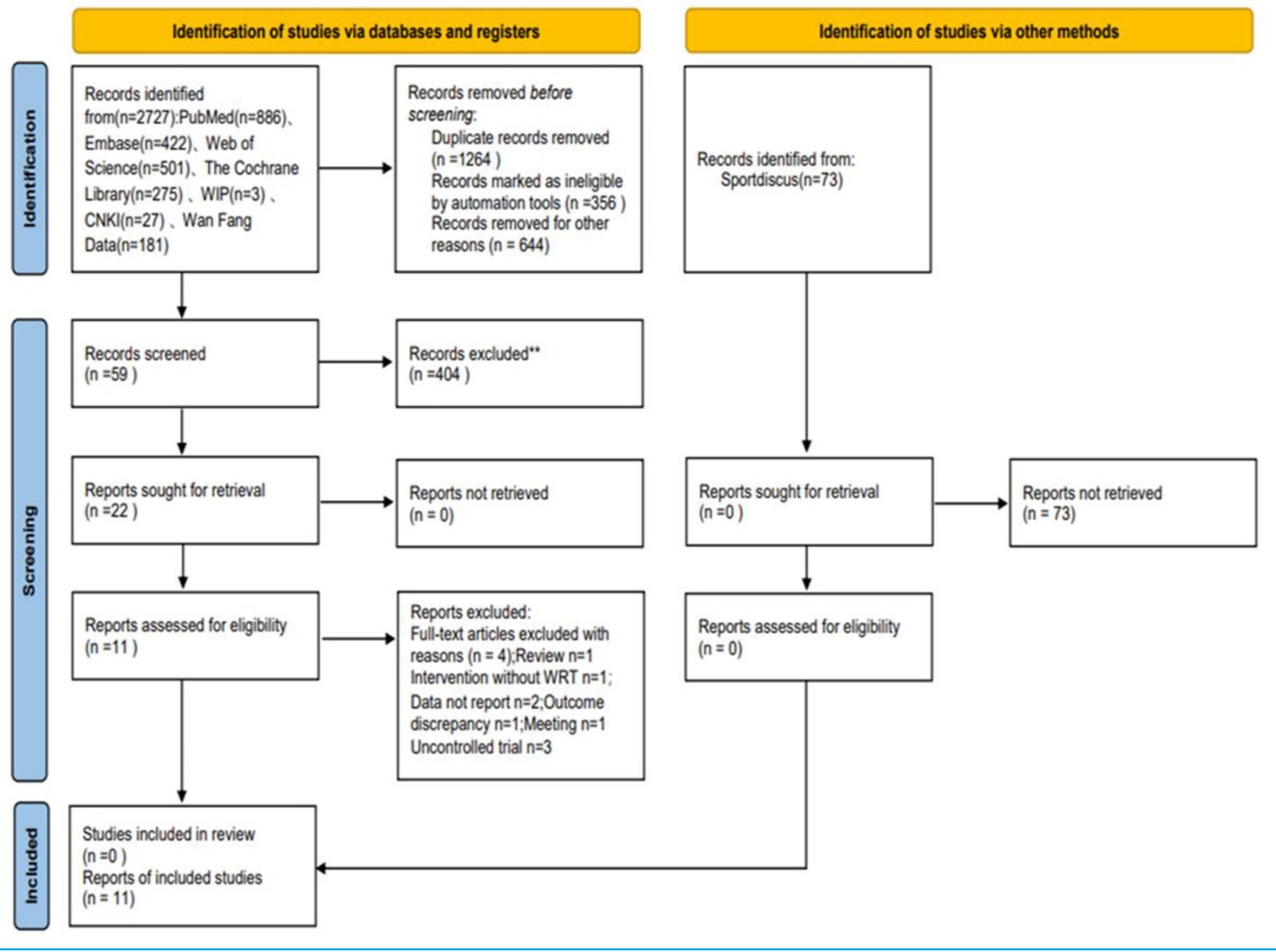

**Figure 1 PRISMA flowchart.**

they were either reviews or theoretical reports. The remaining 59 documents were read, and 11 were finally included according to the inclusion and exclusion criteria. See Fig. 1.

## Basic information on included literature

This study included 11 articles with a total of 1,764 participants. There were 875 participants in the experimental group and 889 in the control group. All participants had type 2 diabetes. The year of publication for the included articles ranged from 2008 to 2024. All the experimental groups in the included articles performed exercise in addition to routine treatment or as the sole treatment. The control groups received routine rehabilitation treatment, diet, or stretching exercises. The included articles contained complete intervention measures, including the intervention period and frequency. Among them: the intervention period for exercise was 4 to 24 weeks; the frequency of intervention

was 3 to 7 times a week; aerobic exercise was used in five articles; yoga exercise was used in four articles; combined exercise was used in four articles; and MDA and SOD, markers of oxidative stress, were used in 13 articles each for assessment. Basic information on the included literature is shown in Table S3.

## Quality assessment of the literature

The 11 included articles were all randomized controlled trials, and all met the criteria of 'similar baseline,' 'intention-to-treat analysis,' 'analysis of statistical results between groups, ' and 'point measurement and difference value' (*Hegde et al., 2020*; *Guo, Bo & Peng, 2014*; *Zhang et al., 2012*; *Promsrisuk et al., 2023*; *Hegde et al., 2011*; *Li, 2013*; *Wang et al., 2015*; *Wei et al., 2023*; *Gordon et al., 2008*; *Vasconcelos Gouveia et al., 2021*; *Xu, Jing & Zhao, 2019*). Two articles met the criteria for 'blinding of therapists' (*Wang et al., 2015*; *Gordon et al., 2008*), and three articles met the criteria for 'blinding of outcome assessment' (*Promsrisuk et al., 2023*; *Gordon et al., 2008*; *Vasconcelos Gouveia et al., 2021*). The PEDro scores ranged from 6 to 8, averaging 6.5. No low-quality literature was found, so the overall quality of the included literature was good. See Table S4.

## Meta-analysis results

### Effect of MDA in patients with Type 2 diabetes

A total of 10 articles reported MDA. The results of the random-effects model meta-analysis are shown in Fig. 2, which shows that the improvement in MDA, an indicator of oxidative stress, in the exercise group of patients with type 2 diabetes was significantly lower than that in the control group. The difference was statistically significant [SMD = −1.29, 95% CI [−1.87 to −0.71], $P < 0.001$]. See Fig. 2.

### Effects of SOD in patients with Type 2 diabetes

A total of seven articles reported on SOD. The results of the random-effects model meta-analysis are shown in Fig. 3, which shows that the improvement in SOD in patients with type 2 diabetes in the exercise group was significantly higher than that in the control group, and the difference was statistically significant [SMD = 0.59, 95% CI [0.17 to 1.01], $P = 0.006$]. See Fig. 3.

### Subgroup analysis

In order to explore possible sources of heterogeneity, this article performs subgroup analyses of MDA and SOD, indicators of oxidative stress, as shown in Tables 1 and 2. The effect of exercise on MDA and SOD, indicators of oxidative stress, in patients with type 2 diabetes may be affected by factors such as age, intervention method, intervention period, and frequency.

### Subgroup analysis of MDA, a marker of oxidative stress

Among the MDA indicators, the age groups included in the study can be divided into two subgroups: 45 to 60 years old and over 60 years old; in terms of intervention methods, they can be divided into four subgroups: aerobic, combined, resistance, and mind-body exercises; in terms of intervention cycle, they can be divided into ≤8 weeks, 8 to 12 weeks,

| Study or Subgroup | Experimental Mean | SD | Total | Control Mean | SD | Total | Weight | Std. Mean Difference IV, Random, 95% CI |
|---|---|---|---|---|---|---|---|---|
| Jiao Wei2023(1) | -46.08 | 5.05 | 40 | -41.23 | 6.37 | 40 | 7.3% | -0.84 [-1.29, -0.38] |
| Jiao Wei2023(2) | -45.85 | 5.12 | 40 | -41.23 | 6.37 | 40 | 7.3% | -0.79 [-1.25, -0.34] |
| Jiao Wei2023(3) | -53.16 | 5.67 | 40 | -41.23 | 6.37 | 40 | 7.1% | -1.96 [-2.50, -1.42] |
| Lorenzo A Gordon2008(1) | -0.47 | 0.183303 | 77 | 0.02 | 0.1253 | 77 | 7.3% | -3.11 [-3.58, -2.63] |
| Lorenzo A Gordon2008(2) | -0.42 | 0.111355 | 77 | 0.02 | 0.1253 | 77 | 7.2% | -3.69 [-4.22, -3.17] |
| Samara Sousa Vasconcelos Gouveia2021 | -3.24 | 5.586779 | 22 | 1.85 | 7.55143 | 22 | 7.0% | -0.75 [-1.37, -0.14] |
| SHREELAXMI V. HEGDE, PHD2011 | -10.8 | 10.66911 | 60 | 1.6 | 10.19461 | 63 | 7.4% | -1.18 [-1.57, -0.80] |
| Shreelaxmi V. Hegde2020 | -4.9 | 8.825531 | 14 | 1.6 | 4.911212 | 15 | 6.7% | -0.89 [-1.66, -0.12] |
| Tichanon Promsrisuk2023 | -0.46 | 0.875043 | 21 | 0.01 | 1.126632 | 21 | 7.0% | -0.46 [-1.07, 0.16] |
| Xiaobing Li2012 | -1.28 | 1.375245 | 30 | -0.76 | 1.52915 | 30 | 7.2% | -0.35 [-0.86, 0.16] |
| Yanli Guo2014 | -0.5 | 0.3 | 20 | 0.1 | 0.360555 | 20 | 6.8% | -1.77 [-2.52, -1.03] |
| Yan Zhang2012 | 0.18 | 1.807733 | 50 | -0.15 | 1.627237 | 49 | 7.4% | 0.19 [-0.20, 0.59] |
| Yuxin Xu2019 (1) | -2.4 | 2.072462 | 38 | -0.88 | 1.859543 | 36 | 7.2% | -0.76 [-1.24, -0.29] |
| Yuxin Xu2019 (2) | -4.08 | 1.965019 | 42 | -0.88 | 1.859543 | 36 | 7.2% | -1.65 [-2.17, -1.13] |
| **Total (95% CI)** | | | 571 | | | 566 | 100.0% | **-1.29 [-1.87, -0.71]** |

Heterogeneity: Tau² = 1.15; Chi² = 237.57, df = 13 (P < 0.00001); I² = 95%
Test for overall effect: Z = 4.34 (P < 0.0001)

**Figure 2** MDA forest diagram (*Wei et al., 2023*; *Gordon et al., 2008*; *Vasconcelos Gouveia et al., 2021*; *Hegde et al., 2011*; *Promsrisuk et al., 2023*; *Li, 2013*; *Guo, Bo & Peng, 2014*; *Zhang et al., 2012*; *Xu, Jing & Zhao, 2019*).

| Study or Subgroup | Experimental Mean | SD | Total | Control Mean | SD | Total | Weight | Std. Mean Difference IV, Random, 95% CI |
|---|---|---|---|---|---|---|---|---|
| Jie Wang2015 (1) | 7.26 | 13.85939 | 41 | -0.45 | 14.57915 | 46 | 10.6% | 0.54 [0.11, 0.97] |
| Jie Wang2015 (2) | 6.56 | 13.01448 | 37 | -0.45 | 14.57915 | 46 | 10.5% | 0.50 [0.06, 0.94] |
| Jie Wang2015 (3) | 6.31 | 13.68222 | 43 | -0.45 | 14.57915 | 46 | 10.6% | 0.47 [0.05, 0.90] |
| SHREELAXMI V. HEGDE, PHD2011 | 6.31 | 20.0476 | 60 | 1.87 | 23.22568 | 63 | 11.0% | 0.20 [-0.15, 0.56] |
| Shreelaxmi V. Hegde2020 | -5.72 | 17.2138 | 14 | 12.66 | 15.76037 | 15 | 8.4% | -1.08 [-1.87, -0.30] |
| Tichanon Promsrisuk2023 | 2.45 | 18.72353 | 21 | -0.8 | 12.55954 | 21 | 9.5% | 0.20 [-0.41, 0.81] |
| Xiaobing Li2012 | 18.83 | 16.39059 | 30 | 8.02 | 16.37321 | 30 | 10.1% | 0.65 [0.13, 1.17] |
| Yanli Guo2014 | 0.5 | 0.360555 | 20 | 0.1 | 0.52915 | 20 | 9.3% | 0.87 [0.21, 1.52] |
| Yuxin Xu(2) | 18.57 | 8.413614 | 42 | 1.89 | 5.00137 | 36 | 9.7% | 2.34 [1.76, 2.93] |
| Yuxin Xu2019 (1) | 9.3 | 8.210018 | 38 | 1.89 | 5.00137 | 36 | 10.3% | 1.07 [0.58, 1.56] |
| **Total (95% CI)** | | | 346 | | | 359 | 100.0% | **0.59 [0.17, 1.01]** |

Heterogeneity: Tau² = 0.38; Chi² = 63.15, df = 9 (P < 0.00001); I² = 86%
Test for overall effect: Z = 2.77 (P = 0.006)

**Figure 3** SOD forest diagram (*Wang et al., 2015*; *Hegde et al., 2011*, *2020*; *Promsrisuk et al., 2023*; *Li, 2013*; *Guo, Bo & Peng, 2014*; *Xu, Jing & Zhao, 2019*).

**Table 1 Results of the meta-analysis of the effect of exercise on MDA, an indicator of oxidative stress, in patients with type 2 diabetes.**

| Outcome indicator | | Number of studies included | P-value | $I^2$% | Meta-analysis results | |
|---|---|---|---|---|---|---|
| | | | | | SMD (95% CI) | P-value |
| MDA | | 10 (1,337) | | 95 | −1.29 [−1.87 to −0.71] | <0.0001 |
| Age | 45–59 years | 7 (591) | 0.18 | 87 | −0.93 [−1.43 to −0.43] | 0.0003 |
| | 60 years and over | 7 (546) | | 96 | −1.74 [−2.84 to −0.65] | 0.0020 |
| Type of intervention | Aerobics | 5 (447) | 0.55 | 97 | −1.37 [−2.69 to −0.09] | 0.040 |
| | Joint movement | 3 (200) | | 86 | −1.37 [−2.22 to −0.52] | 0.0020 |
| | Movement of body and mind | 5 (410) | | 95 | −1.27 [−2.27 to −0.27] | 0.010 |
| Intervention period | ≤8 weeks | 6 (443) | 0.10 | 88 | −0.74 [−1.32 to −0.16] | 0.010 |
| | 8~12 weeks | 4 (234) | | 62 | −1.07 [−1.55 to −0.58] | <0.0001 |
| | >12 weeks | 4 (460) | | 96 | −2.30 [−3.62 to −0.98] | 0.0006 |
| Frequency of intervention | <3 d/week | 2 (308) | <0.0001 | 63 | −3.39 [−3.96 to −2.81] | <0.0001 |
| | 3–5 d/week | 7 (485) | | 87 | −0.91 [−1.46 to −0.37] | 0.0010 |
| | >5 d/week | 5 (344) | | 80 | −0.93 [−1.44 to −0.43] | 0.0003 |

Note:

SMD, Standard Mean Difference; $I_2$, Heterogeneity test; d/week, day/week.

**Table 2 Results of the meta-analysis of the effect of exercise on SOD, an indicator of oxidative stress, in patients with type 2 diabetes.**

| Outcome indicator | | Number of studies included | P-value | $I^2$% | Meta-analysis results | |
|---|---|---|---|---|---|---|
| | | | | | SMD (95% CI) | P-value |
| SOD | | 7 (705) | | 69 | 0.59 [0.17–1.01] | 0.006 |
| Age | 45–59 years | 4 (252) | 0.30 | 82 | 0.20 [−0.45 to 0.85] | 0.540 |
| | 60 years and over | 5 (375) | | 31 | 0.57 [0.32–0.82] | <0.001 |
| Type of intervention | Aerobics | 2 (114) | 0.01 | 0 | 1.00 [0.61–1.39] | <0.001 |
| | Movement of body and mind | 6 (471) | | 69 | 0.30 [−0.05 to 0.64] | 0.090 |
| Intervention period | ≤12 weeks | 5 (294) | 0.16 | 76 | 0.21 [−0.30 to 0.72] | 0.420 |
| | >12 weeks | 4 (333) | | 28 | 0.63 [0.36–0.89] | <0.001 |
| Frequency of intervention | <5 d/week | 5 (422) | 0.65 | 0 | 0.45 [0.25–0.64] | <0.001 |
| | ≥5 d/week | 4 (205) | | 86 | 0.26 [−0.54 to 1.05] | 0.530 |

Note:

SMD, Standard Mean Difference; $I_2$, Heterogeneity test; d/week, day/week.

and >12 weeks; in terms of intervention frequency, they can be divided into three subgroups: ≤3 d/week, 3 to 5 d/week, and >5 d/week.

The results of the subgroup analysis (Table 1) showed that all differences were statistically significant. When the type 2 diabetes patients were aged over 60 years [$I^2$ = 96%, SMD = −1.74, 95% CI [−2.84 to −0.65], $P$ = 0.002], the intervention method was aerobic exercise [$I^2$ = 97%, SMD = −1.37, 95% CI [−2.69 to −0.09], $P$ = 0.04] and combined exercise [$I^2$ = 86%, SMD = −1.37, 95% CI [−2.22 to −0.52], $P$ = 0.002], intervention duration >12 weeks [$I^2$ = 96%, SMD = −2.30, 95% CI [−3.62 to −0.98], $P$ = 0.0006] and intervention frequency <3 d/week [$I^2$ = 63%, SMD = −3.39, 95% CI [−3.96 to −2.81], $P$ < 0.0001], the effect of exercise on MDA, an indicator of oxidative stress, was more significant in patients with type 2 diabetes. There was no significant change in the effect

heterogeneity of age, intervention method, intervention cycle, and intervention frequency, indicating that age, intervention method, intervention cycle, and intervention frequency may not be the source of heterogeneity.

This indicator is included in resistance training. This study shows that resistance training can improve MDA, an oxidative stress marker, in patients with type 2 diabetes. Therefore, it will not be quantitatively analyzed but only descriptively analyzed.

*Subgroup analysis of SOD, a marker of oxidative stress*

Among the SOD indicators, the age groups included in the study can be divided into two subgroups: 45–60 years old and over 60 years old; in terms of intervention methods, they can be divided into three subgroups: aerobic, combined, and physical and mental exercises; in terms of intervention period, they can be divided into ≤12 weeks and >12 weeks; in terms of intervention frequency, they can be divided into two subgroups: <5 d/week and ≥ 5d/week.

The results of the subgroup analysis (Table 2) showed that when the intervention method was physical and mental exercise [$I^2$ = 69%, SMD = 0.30, 95% CI [−0.05 to 0.64], $P$ = 0.09], the age was 45–59 years [$I^2$ = 82%, SMD = 0.20, 95% CI [−0.45 to 0.85], $P$ = 0.54], intervention period ≤12 weeks [$I^2$ = 76%, SMD = 0.21, 95% CI [−0.30 to 0.72], $P$ = 0.42], and frequency ≥ 5 days/week [$I^2$ = 86%, SMD = 0.26, 95% CI [−0.54 to 1.05], $P$ = 0.53], the differences were not statistically significant. When the participants with type 2 diabetes were aged over 60 [$I^2$ = 31%, SMD = 0.57, 95% CI [0.32–0.82], $P$ < 0.0001], the intervention method was aerobic exercise [$I^2$ = 0%, SMD = 1.0 0, 95% CI [0.61–1.39], $P$ < 0.001], intervention duration >12 weeks [$I^2$ = 28%, SMD = 0.63, 95% CI [0.36–0.89], $P$ < 0.001] and intervention frequency <5 d/week [$I^2$ = 0%, SMD = 0.45, 95% CI [0.25–0.64], $P$ < 0.0001], the effect of exercise on SOD, an oxidative stress marker in patients with type 2 diabetes, was more significant. In terms of the source of heterogeneity, the effect heterogeneity of age, intervention method, intervention cycle, and intervention frequency is less than 50%, and the heterogeneity is significantly reduced. It can be seen that age, intervention method, intervention cycle, and intervention frequency may be the source of heterogeneity.

This indicator also incorporates joint training. This study shows that joint training can also improve SOD, an oxidative stress marker, in patients with type 2 diabetes. Therefore, it will not be quantitatively analyzed but only descriptively analyzed.

### Sensitivity analysis

To explore whether a single study caused the heterogeneity between articles, this study performed a sensitivity analysis of the research on the effect of exercise on MDA and SOD, two indicators of oxidative stress, in patients with type 2 diabetes. A sensitivity analysis was performed by excluding each study one by one and combining the effects. See Table 3. After excluding the study by *Gordon et al. (2008)*, the combined effect of the oxidative stress marker MDA was SMD = −0.92, 95% CI [−1.29 to −0.56], $P$ < 0.0001; $I^2$ decreased from 95% to 83%, and there was still high heterogeneity; after excluding the articles by *Gordon et al. (2008)* and *Zhang et al. (2012)*, the combined effect of the oxidative stress

**Table 3 Removal of the combined effect of the individual study oxidative stress markers MDA and SOD.**

| | Inclusion of studies | Effective quantity | 95% CI | P (Merger effects) | $I^2$% |
|---|---|---|---|---|---|
| | Hegde et al. (2011) | −1.29 | [−1.94 to −0.65] | <0.001 | 95 |
| | Guo, Bo & Peng (2014) | −1.25 | [−1.86 to −0.64] | <0.001 | 95 |
| | Zhang et al. (2012) | −1.40 | [−1.97 to −0.84] | <0.001 | 93 |
| | Promsrisuk et al. (2023) | −1.35 | [−1.96 to −0.74] | <0.001 | 95 |
| | Hegde et al. (2020) | −1.31 | [−1.93 to −0.70] | <0.001 | 95 |
| MDA | Li (2013) | −1.36 | [−1.97 to −0.75] | <0.001 | 95 |
| | Wei et al. (2023) (1) | −1.32 | [−1.95 to −0.69] | <0.001 | 95 |
| | Wei et al. (2023) (2) | −1.32 | [−1.95 to −0.70] | <0.001 | 95 |
| | Wei et al. (2023) (3) | −1.23 | [−1.85 to −0.62] | <0.001 | 95 |
| | Gordon et al. (2008) (1) | −1.14 | [−1.68 to −0.61] | <0.001 | 93 |
| | Gordon et al. (2008) (2) | −1.10 | [−1.59 to −0.61] | <0.001 | 92 |
| | Vasconcelos Gouveia et al. (2021) | −1.33 | [−1.94 to −0.71] | <0.001 | 95 |
| | Xu, Jing & Zhao (2019) (1) | −1.33 | [−1.95 to −0.70] | <0.001 | 95 |
| | Xu, Jing & Zhao (2019) (2) | −1.26 | [−1.88 to −0.64] | <0.001 | 95 |
| | Hegde et al. (2011) | 0.64 | [0.17–1.11] | 0.008 | 86 |
| | Guo, Bo & Peng (2014) | 0.56 | [0.11–1.02] | 0.010 | 87 |
| | Promsrisuk et al. (2023) | 0.63 | [0.18–1.09] | 0.010 | 87 |
| | Hegde et al. (2020) | 0.74 | [0.36–1.12] | 0.001 | 82 |
| | Li (2013) | 0.59 | [0.12–1.05] | 0.010 | 87 |
| SOD | Wang et al. (2015) (1) | 0.60 | [0.12–1.08] | 0.010 | 87 |
| | Wang et al. (2015) (2) | 0.60 | [0.13–1.08] | 0.010 | 87 |
| | Wang et al. (2015) (3) | 0.61 | [0.13–1.08] | 0.010 | 87 |
| | Xu, Jing & Zhao (2019) (1) | 0.54 | [0.08–0.99] | 0.020 | 86 |
| | Xu, Jing & Zhao (2019) (2) | 0.43 | [0.13–0.72] | 0.005 | 69 |

**Note:**
SMD, Standard Mean Difference; $I_2$, Heterogeneity test; 95% CI, 95% Heterogeneity Interval.

marker MDA was The combined effect of MDA, a marker of oxidative stress, was SMD = −1.03, 95% CI [−1.33 to −0.73], $P < 0.0001$; $I^2$ decreased from 95% to 72%, and there was still high heterogeneity; after excluding other individual articles, the range of the combined effect SMD was (−1.10 to −1.40), and the range of $I^2$ was (93% to 95%), both $P < 0.001$.

After excluding the study by *Yuxin, Qingping & Cuihong (2019)*, the pooled effect of SOD, an indicator of oxidative stress, was SMD = 0.43, 95% CI [0.13 – 0.72], $P = 0.005$; $I^2$ decreased from 86 to 69%, and there was still high heterogeneity; after excluding the other individual articles, the range of the combined effect SMD was (0.54 to 0.74), the range of $I^2$ was (82% to 87%), and $P$ was <0.05 (Figs. S1 and S2).

## Publication bias

This study performed a publication bias analysis on the oxidative stress indicators MDA and the outcome indicator SOD. Egger's test for the oxidative stress indicator MDA: P > |t| = 0.7810 > 0.05, oxidative stress indicator SOD: P > |t| = 0.5468 > 0.05. This indicates no publication bias, as shown in Figs. S3 and S4.

## Quality of evidence evaluation

This study included 11 articles with an average PEDro score of 6.5, indicating good quality. Subgroup and sensitivity analyses of the oxidative stress indicator MDA failed to identify sources of heterogeneity, indicating significant limitations. Subgroup and sensitivity analyses of the oxidative stress indicator SOD identified sources of heterogeneity. Publication bias testing for the oxidative stress indicators MDA and SOD suggests that there is no significant publication bias. Therefore, the GRADEPro software shows moderate and high-quality evidence for the oxidative stress indicators MDA and SOD, respectively (Fig. S5).

## Adverse events

None of the 11 included articles reported adverse events caused by exercise.

## DISCUSSION

This study found that exercise can improve MDA, an indicator of oxidative stress, in patients with type 2 diabetes. This result is consistent with previous research (*Venugopal et al., 2022*), suggesting that exercise has a beneficial effect on MDA levels. However, earlier studies did not determine whether exercise could also improve SOD, another oxidative stress marker in type 2 diabetes. Building upon previous research, our study demonstrates that exercise not only improves MDA but also significantly enhances SOD levels. The effect of an exercise intervention on oxidative stress MDA may be better when the intervention method is aerobic exercise and combined exercise; the intervention period is >12 weeks, the intervention frequency is <3 d/week and the age is >60 years. The effect of an exercise intervention on oxidative stress SOD may be better when the intervention method is aerobic exercise, the intervention period is >12 weeks, the intervention frequency is <5 d/week and the age is >60 years. Exercise not only improves insulin sensitivity and blood glucose levels in patients with type 2 diabetes (*Chang, 2011*; *Chen et al., 2024*) but also reduces oxidative stress and lipid peroxidation (*Simioni et al., 2018*). The possible mechanism of action is achieved by reducing the protein expression of NAD (P)H oxidase (*Olesen et al., 2014*). Oxidative stress plays an important role in T2DM. Therefore, improving oxidative stress through exercise can help rehabilitate patients with type 2 diabetes.

This study included 11 articles that systematically reviewed and analysed the intervention effect of exercise on MDA and SOD, indicators of oxidative stress, in patients with type 2 diabetes. The quality of the literature was evaluated using PEDro, with an average score of 6.5. No low-quality literature was found, and the overall quality of the literature was relatively good. Study heterogeneity was a downgrading factor, and most of the literature did not completely report on blinding or implement allocation concealment, which may have an impact on the post-test results. Publication bias tests were performed on the oxidative stress markers MDA and SOD, and the results showed that there was no significant publication bias. No significant reasons for downgrading were found for indirectness and imprecision in the evaluation of the quality of the evidence. The meta-analysis showed that $I^2$ for both oxidative stress indicators MDA and SOD was

greater than 50%, indicating high heterogeneity among articles. Subgroup analysis found that the source of heterogeneity was not found for MDA, but was found for SOD. Sensitivity analysis found that the frequency Therefore, the quality of evidence for the effect of exercise on MDA, an indicator of oxidative stress, in patients with type 2 diabetes is moderate, and the quality of evidence for the effect of exercise on SOD, an indicator of oxidative stress, is high.

Related research has found that 12 weeks of 100 min/week moderate to high-intensity aerobic exercise under medical supervision can reduce HbA1c (*Jayedi, Emadi & Shab-Bidar, 2022*). Combined exercise refers to two or more types of exercise. This article includes aerobic combined resistance exercise, yoga combined resistance exercise, tai chi combined resistance exercise, and other combined exercises. Relevant articles have shown that compared with doing any type of exercise alone, adults with T2DM who receive a combined exercise training program exercise more have a greater reduction in glycosylated hemoglobin A1c (HbA1c) (*Featherstone, Holly & Amsterdam, 1993*), lose more weight, and improve their cardiorespiratory endurance (*Church et al., 2010*). This study found that aerobic exercise had a significant effect on improving SOD, a marker of oxidative stress, compared to mind-body exercise, which had no significant effect. *Venugopal et al. (2022)* also found that yoga had no significant effect on improving SOD, a marker of oxidative stress. The possible reason is that *Venugopal et al. (2022)* only included yoga in their study. This article includes various exercise types: aerobic, mind-body exercise (yoga, tai chi, and others), resistance, and combined exercise. Therefore, the effect of intervention on the oxidative stress marker SOD may be different. This study found that the effect of aerobic exercise on oxidative stress markers SOD was most significant. The possible mechanism is that aerobic exercise can promote the body's utilization of oxygen, improve peripheral insulin sensitivity (*Liu, Xiao & Yang, 2008*), correct abnormal glucose tolerance, activate smooth muscle insulin receptors, enhance insulin binding to receptors, and promote glucose metabolism (*Chen et al., 2019*). Therefore, aerobic exercise should be given priority when choosing an exercise plan. This study also found that the effect of oxidative stress on patients with type 2 diabetes is also affected by the patient's age. When patients with type 2 diabetes are >60 years old, the effect of exercise intervention is better. Type 2 diabetes is more common in middle and old age. With age and various irregular lifestyles, physical function is greatly weakened, and metabolic capacity is reduced, which leads to endocrine disorders and an inability to properly control the metabolic balance of blood sugar in the body. Therefore, age may also be one of the factors affecting oxidative stress.

This study has several limitations. First, because the experimental groups in the included studies underwent combined exercise and drug interventions without isolating these components—where exercise treatments were consistently applied alongside medication—pharmaceutical effects may have confounded the results. Future studies should implement separate controls for exercise and drug interventions to more precisely determine exercise's effects on oxidative stress. Second, despite sensitivity and subgroup analyses, the substantial heterogeneity in MDA (a key oxidative stress marker) persisted, undermining the reliability of the conclusions. Due to this unresolved heterogeneity, the preference for aerobic exercise over mind-body exercises should be interpreted cautiously.

## CONCLUSIONS

Exercise can improve the oxidative stress indicators MDA and SOD in patients with type 2 diabetes. The effect may be affected by the patient's age, method, duration, and frequency of the intervention. This provides evidence-based medical evidence for the clinical prevention and treatment of type 2 diabetes. Current evidence recommends that aerobic exercise and exercise programs for >12 weeks can improve oxidative stress in patients with type 2 diabetes.

### Funding

The authors received no funding for this work.

### Competing Interests

The authors declare that they have no competing interests.

### Author Contributions

- Chen Qiu conceived and designed the experiments, performed the experiments, analyzed the data, prepared figures and/or tables, authored or reviewed drafts of the article, and approved the final draft.
- Shufan Li conceived and designed the experiments, performed the experiments, analyzed the data, authored or reviewed drafts of the article, and approved the final draft.
- Shuqi Jia conceived and designed the experiments, performed the experiments, analyzed the data, authored or reviewed drafts of the article, and approved the final draft.
- Fen Yu conceived and designed the experiments, performed the experiments, analyzed the data, authored or reviewed drafts of the article, and approved the final draft.
- Chen Wei conceived and designed the experiments, authored or reviewed drafts of the article, and approved the final draft.
- Xing Wang conceived and designed the experiments, authored or reviewed drafts of the article, and approved the final draft.
- Bo Guo conceived and designed the experiments, authored or reviewed drafts of the article, and approved the final draft.

### Data Availability

This is a systematic review and meta-analysis.

### Supplemental Information

Supplemental information for this article can be found online at http://dx.doi.org/10.7717/peerj.19814#supplemental-information.

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
