# Peer review of "The effects of exercise on oxidative stress MDA and SOD in patients with type 2 diabetes: a systematic review and meta-analysis"

_PeerJ, doi:10.7717/peerj.19814_

## Round 0.1 · original submission · Major Revisions

Dear Dr. Qiu,

Your manuscript entitled “The effects of exercise on oxidative stress in patients with type 2 diabetes: a systematic review and meta-analysis", which you submitted to PeerJ, has been reviewed by the editor and two experts in the field.

The reviewers generally support your work but have raised significant concerns that must be addressed before the manuscript can move forward. I would happily reconsider your manuscript if you undertake these substantial revisions and resubmit.

If you decide to resubmit the revised version, please summarize all the improvements made in the new version and give answers to all critical points raised in the reviewers’ report in an accompanying letter. Copy and paste each and every reviewer's comment above your response.

Please note that resubmitting your manuscript does not guarantee eventual acceptance. Since the requested changes are major, the revised manuscript will undergo a second round of review by the same reviewers. I must emphasize that the acceptability of the revision will depend upon the resolution of the points raised by the reviewers.

Sincerely yours,

Stefano Menini

Reviewer 1 ·

Basic reporting

-

Experimental design

Authors need to outline clearly some of the methods followed

Validity of the findings

-

Additional comments

The authors have conducted a meta-analysis exploring the effect of exercise on oxidative stress in individuals with type 2 diabetes mellitus (T2DM). While the work is interesting and relevant, I have provided several comments aimed at improving the overall quality of the manuscript:
Abstract
1. Line 44: Include T2DM as the abbreviation for type 2 diabetes mellitus.
2. Lines 52–53: Revise the sentence to read: "...can improve MDA, SMD = -1.29, 95% CI (-1.87, -0.71), p < 0.0001 in patients with T2DM." Apply this format consistently to all other reported biomarkers.

Introduction
1. Authors should define key terms upon first use and provide their abbreviations, which should then be used consistently throughout the text. For instance, define type 2 diabetes mellitus (T2DM) early in the introduction.
2. The references do not follow chronological order. For example, reference 39 appears before reference 3, and reference 35 appears before 4. This issue is observed throughout the reference list—please revise to ensure proper sequencing.
3. Line 106: Rephrase to: "easy to operate and organize."
4. Although the manuscript focuses on exercise as a therapeutic intervention, it is important to briefly mention other treatments that have been explored for oxidative stress in diabetes to better motivate the choice of exercise. For instance, pomegranate has recently been reported to possess antioxidative properties in diabetes (see: https://doi.org/10.3390/antiox12081566). You may use this to support your rationale and perhaps discuss limitations of such alternatives, especially in relation to inflammation.
Methods
1. Reference 26 (PRISMA): This reference is incorrect. The current citation is:
Qiu Zhuoying, Li Lun, Chen Di, et al. A study of the World Health Organisation International Classification of Health-based Family Rehabilitation Guidelines: theoretical framework and methodological system. Chinese Rehabilitation Theory and Practice, 2020, 26(02): 125–3.
Please amend this and cite the appropriate PRISMA guideline.
2. Line 143: Provide the exact date in the format dd mm yyyy.
3. Line 151: Indicate the version and provide a citation for the EndNote software used.
4. Line 184: Authors mention using the PEDro scale for quality assessment; however, the reference provided (ref 26) is incorrect. Please replace it with the correct source.
5. Line 192: The GRADEpro tool is incorrectly referenced. Reference 28 in the list does not relate to evidence grading. Please correct this and cite the proper GRADE guideline.
6. There is inconsistent use of terms such as papers, articles, and studies. Choose one term and apply it consistently throughout the manuscript.

Results
1. All figures must be improved for clarity. The current images are blurry and not reader-friendly.
2. Be specific in identifying groups within your study. Clearly state which groups were considered experimental and which were control.

·

Basic reporting

General Comments
The present systematic review and meta-analysis aimed to investigate the effect of exercise on oxidative stress (as measured by MDA and SOD) in patients with type 2 diabetes. The topic is relevant and clinically meaningful, especially as non-pharmacological interventions gain interest. The manuscript presents several strengths, including a registered protocol, adherence to PRISMA guidelines, and use of validated tools (PEDro, GRADEPro). However, several sections of the manuscript require improvements for clarity, methodological transparency, and scientific rigor.

Specific Comments
Please consider the following point-by-point revisions:

Title and Abstract
• Title: The title is clear but could be more informative by highlighting that this is a meta-analysis on biomarkers of oxidative stress (e.g., “malondialdehyde and superoxide dismutase”).
• Lines 42–77 (Abstract): The abstract is overloaded with statistical data but lacks clarity in structure. Please present the objectives, methods, main findings (with fewer details), and conclusion in a more concise and reader-friendly manner.
• Lines 58–66: Subgroup analysis details should be trimmed and key findings summarized.
• Lines 68–77: Add a sentence on the practical implications and future research directions.

Introduction
The introduction covers key background aspects, but it needs refinement in focus and citations.
• Lines 87–91: Please update prevalence data using the most recent IDF report.
• Lines 106–121: Clarify the mechanisms by which exercise may reduce oxidative stress and cite more diverse studies, including international evidence.
• Lines 122–133: The rationale for the review is sound. However, the aim could be rewritten using narrative language to enhance clarity (e.g., “This study aimed to...” instead of “Therefore, this study intends to...”)

Experimental design

Methods
The methods section is mostly adequate but lacks some transparency.
• Lines 141–149 (Search strategy): Please provide an exact search string used in one major database (e.g., PubMed) in the supplementary material.
• Lines 159–167 (Inclusion criteria): Please clarify the age threshold (“g45 years old” is unclear—do you mean ≥45?).
• Line 183–191 (PEDro assessment): Was there an inter-rater agreement analysis? Please clarify how disagreements were resolved.
• Lines 192–195 (GRADE): Specify what domains were downgraded (risk of bias, inconsistency?) for each outcome.
• Line 204 (Meta-analysis): The cutoff for heterogeneity is standard, but please report whether fixed or random effects were used per outcome explicitly.
• Lines 295–311 (Sensitivity analysis): Consider presenting sensitivity plots or influence diagnostics in the supplementary material.

Validity of the findings

Results
• Line 209–213: Please report the number of studies excluded at each stage in the PRISMA flow diagram and whether any were excluded due to language.
• Tables 1–3: Tables are informative but need clear legends (e.g., explain abbreviations like SMD, I², d/week).
• Line 257: For subgroup analyses, please provide between-group p-values to evaluate interaction effects.
• Line 268: The decision to descriptively analyze some indicators should be justified (e.g., why was resistance training excluded from meta-analysis?).
• Line 294–311: Sensitivity analysis suggests robustness, but the residual heterogeneity remains high—please comment more directly on this in the discussion.


Discussion and Conclusion
• Lines 327–381: The discussion is generally well developed. However, the mechanism linking exercise to reduced oxidative stress should be discussed more critically.
• Lines 356–375: Provide a comparative summary with other recent systematic reviews or meta-analyses (e.g., Venugopal et al. 2022).
• Lines 382–387: The limitation about co-intervention with drugs is valid. Please consider proposing strategies to isolate exercise effects in future research.
• Lines 388–393: The conclusions are supported by the data. However, the recommendation to prefer aerobic exercise over mind-body interventions should be made cautiously due to study heterogeneity.

References
• The reference list is comprehensive and includes many recent sources.
• Please ensure consistency in formatting (e.g., line 425 contains an abbreviated title; some use first author only, others use multiple names).

Additional comments

Check figures quality. They were wrongly adjusted to the page.

---

## Round 0.2 · Minor Revisions

Please address this remaining point from Reviewer 1. Feel free to include suggested references only if you believe they add value or relevance. My decision will consider your reasoning for including or not including them, rather than whether every reviewer's request is followed
Sincerely,
Stefano Menini

**PeerJ Staff Note**: Please ensure that all review, editorial, and staff comments are addressed in a response letter and that any edits or clarifications mentioned in the letter are also inserted into the revised manuscript where appropriate.

**PeerJ Staff Note**: It is PeerJ policy that additional references suggested during the peer-review process should only be included if the authors agree that they are relevant and useful.

Reviewer 1 ·

Basic reporting

Thanks to the authors for revising the previous version; it has substantially improved.
However, not all comments have been addressed
"Although the manuscript focuses on exercise as a therapeutic intervention, it is important to briefly mention other treatments that have been explored for oxidative stress in diabetes to better motivate the choice of exercise. For instance, pomegranate has recently been reported to possess antioxidative properties in diabetes (see: https://doi.org/10.3390/antiox12081566). You may use this to support your rationale and perhaps discuss limitations of such alternatives, especially in relation to inflammation."
Incorporate this recent study in the introduction along lines 131-135. Note, while you focus on exercise, it's important to state why exercise, as other alternative therapies are available that also improve oxidative stress in type 2 diabetes. The argument here can be that the evidence on pomegranate is primarily in preclinical studies, which are difficult to translate in clinical settings, hence prompting authors to focus on exercise.

Experimental design

Fine

Validity of the findings

Fine

Additional comments

Not applicable

·

Basic reporting

The authors adressed all my concerns

Experimental design

The authors adressed all my concerns

Validity of the findings

The authors adressed all my concerns

Additional comments

The authors adressed all my concerns

---

## Round 0.3 · accepted · Accept

Dear Dr. Qiu,

Thank you for submitting the revised version of your manuscript. I have personally reviewed the revision and confirmed that all the reviewers' comments have been adequately addressed. The quality of the manuscript has significantly improved as a result. I am pleased to inform you that your manuscript is now ready for publication in PeerJ in its current form.

Sincerely yours,
Stefano Menini